# Bioluminescence Production by Turnip Yellows Virus Infectious Clones: A New Way to Monitor Plant Virus Infection

**DOI:** 10.3390/ijms232213685

**Published:** 2022-11-08

**Authors:** Sylvaine Boissinot, Marie Ducousso, Véronique Brault, Martin Drucker

**Affiliations:** 1Santé de la Vigne et Qualiité du Vin, Unité Mixte de Recherche 1131, Institut National de Recherche pour l’Agriculture, l’Alimentation et l’Environnement, Centre Grand Est, Université Strasbourg, 68000 Colmar, France; 2Plant Health Institute Montpellier, Institut National de Recherche pour l’Agriculture, l’Alimentation et l’Environnement, CIRAD, Institut de Recherche pour le Développement, Institut Agro, Université Montpellier, 34980 Montferrier sur Lez, France

**Keywords:** turnip yellows virus, host plant, bioluminescence, in vivo imaging

## Abstract

We used the NanoLuc luciferase bioluminescent reporter system to detect turnip yellows virus (TuYV) in infected plants. For this, TuYV was genetically tagged by replacing the C-terminal part of the RT protein with full-length NanoLuc (TuYV-NL) or with the N-terminal domain of split NanoLuc (TuYV-N65-NL). Wild-type and recombinant viruses were agro-infiltrated in *Nicotiana benthamiana*, *Montia perfoliata*, and *Arabidopsis thaliana*. ELISA confirmed systemic infection and similar accumulation of the recombinant viruses in *N. benthamiana* and *M. perfoliata* but reduced systemic infection and lower accumulation in *A. thaliana*. RT-PCR analysis indicated that the recombinant sequences were stable in *N. benthamiana* and *M. perfoliata* but not in *A. thaliana*. Bioluminescence imaging detected TuYV-NL in inoculated and systemically infected leaves. For the detection of split NanoLuc, we constructed transgenic *N. benthamiana* plants expressing the C-terminal domain of split NanoLuc. Bioluminescence imaging of these plants after agro-infiltration with TuYV-N65-NL allowed the detection of the virus in systemically infected leaves. Taken together, our results show that NanoLuc luciferase can be used to monitor infection with TuYV.

## 1. Introduction

Following virus infection in real-time and in a non-destructive manner is desirable for various reasons, and different techniques have been developed or adapted for this. Most approaches rely on inserting a marker into the viral genome that can be followed easily in infected organisms. Each marker has its advantages and inconveniences. For example, the recombinant turnip yellows virus (TuYV) TuM1s81 contains a hairpin insertion for silencing the *Arabidopsis thaliana CHLI1* gene [1]. This allows easy visualization of the otherwise symptomless viral infection by bleaching the infected tissue. However, observation is only possible at the macroscopic scale and restricted to photosynthetic tissue and to *A. thaliana*; observation in roots is not possible. Similar restrictions applied to an approach where a transcription factor was inserted into the genome of the tobacco etch virus that induced the accumulation of colored anthocyanins in infected tissue [2]. More popular markers used to follow virus infection are fluorescent proteins that are inserted in the viral genome to be expressed as a free form or fused to a viral protein [3,4,5]. Macroscopic or microscopic observation of fluorescent proteins requires equipment available in most laboratories and does not require specific sample preparation. Drawbacks can be high autofluorescence of plant tissue, bleaching and restriction of non-destructive microscopic observation to the superficial tissue layers, and visualization in deeper tissues such as the phloem requiring peeling of the epidermis or sectioning. These limitations do not apply to bioluminescence approaches, where a luciferase is inserted into a virus genome that emits light after the addition of specific substrates. In general, this technique offers high signal-to-noise ratios and a high linear signal range, allowing quantification. Autofluorescence is no issue, and non-destructive observation deep in the tissue is possible. However, infiltration of luciferase substrate and the need for darkroom conditions can pose a problem. For this reason, bioluminescence is mainly used for in vitro measurements; macroscopic and microscopic observations are less common. Luciferase reporter viruses have been designed mostly for animal and human viruses, for example, hepatitis C virus, [6], herpes simplex virus 1 [7], HIV-1 [8], or dengue virus [9]. Examples of luciferase-tagged plant viruses are scarce. A Pubmed research (https://pubmed.ncbi.nlm.nih.gov/?term=plant+virus+luciferase+reporter, accessed on 18 August 2022) using the terms “plant virus luciferase reporter” identified only one report on a luciferase-tagged plant virus: potato virus A [10]. All other studies were either unrelated or used bioluminescence to characterize plant virus promoters.

A common problem of inserting reporter genes in viral genomes is genome instability. Usually, the insertion does not present a selective advantage for the virus, favoring its removal [11,12]. Reducing the size of the reporter can increase its stability in the virus genome. For bioluminescent reporters, there is—besides the commonly used firefly and *Renilla* luciferases (61 and 36 kDa, respectively)—a 19 kDa NanoLuc luciferase available, which is a genetically modified version of a bioluminescent enzyme originally isolated from the deep sea shrimp *Oplophorus gracilirostris* [13]. Besides its smaller size, it is about 150 times more active than the firefly and *Renilla* luciferases [13], making it an attractive alternative for creating a bioluminescent reporter virus. A split version of NanoLuc is also available, comprising a 7.2 kDa N-terminal and an 11.8 kDa C-terminal fragment that can self-assemble to reconstitute a functional NanoLuc [14]. This further reduces the size of the reporter gene to be inserted but requires providing the other fragment in trans.

We investigated here whether NanoLuc or split Nanoluc can be applied to study infection with a plant virus. For this, we used turnip yellows virus (TuYV, genus *Polerovirus*, family *Solemoviridae*). TuYV is an icosahedral 5.6 kbp single-stranded positive-sense RNA virus that infects multiple plant species, amongst them the model plants *Arabidopsis thaliana* and *Nicotiana benthamiana*. Its single RNA has a 5′ VPg and no 3′-poly(A) tail. It codes like other poleroviruses for about 10 proteins, among them the RNA silencing suppressor P0, an RNA-dependent replicase, two movement proteins, and a capsid protein as well as its extension, and the readthrough protein (RT) that is translated by a readthrough mechanism of the CP stop codon. The CP and a truncated form of the RT (RT*) are incorporated into the capsid. TuYV is phloem-restricted and transmitted by aphids (https://ictv.global/report/chapter/solemoviridae/solemoviridae/polerovirus, accessed 24 October 2022).

## 2. Results

### 2.1. Infectivity of TuYV-NL and TuYV-N65-NL in Agro-Inoculated Leaves and in Systemic Leaves

In order to evaluate the use of NanoLuc and split NanoLuc luciferase to monitor virus infection in plants, two TuYV mutants were generated by modification of the C-terminal part of the readthrough protein (RT) in a TuYV infectious clone [15]. The C-terminus of the TuYV RT protein is mandatory for virus transmission by aphids but not required for systemic propagation in plants [16,17,18]. Replacement of the C-terminal part of the RT protein by the GFP sequence has already been successfully used to produce a TuYV clone that can establish a systemic infection in plants and that can be visualized by real-time fluorescence microscopy in inoculated plants [19]. Using the same strategy, we replaced here the C-terminal 171 amino acids of the RT protein with the full-length NanoLuc (NL) sequence in the TuYV-NL mutant (Figure 1). For the split NanoLuc strategy, the C-terminal 65 amino acids of the RT protein were replaced by the N-terminal 65 amino acids of the NanoLuc protein (TuYV-N65-NL, Figure 1). This resulted in mutants having the same genome length as wild-type TuYV, but with different deletions of the 3′ extremity of the RT sequence (513 nucleotides for TuYV-NL and 195 nucleotides for TuYV-N65-NL).

We agro-infiltrated *Montia perfoliata* and *Nicotiana benthamiana* leaves with the wild-type and mutant constructs to assess the infectivity of the virus mutants. Western blots of total extracts prepared from agro-infiltrated tissues showed that CP (22 kDa) accumulated in leaves agro-inoculated with mutants TuYV-NL and TuYV-N65 to similar levels as in leaves inoculated with wild-type TuYV (Figure 2). This was true for both agro-infiltrated *M. perfoliata* and *N. benthamiana* leaves, indicating that the RT sequence modifications did not affect virus replication and synthesis of the subgenomic mRNA1 [21] from which CP is translated. We also looked for western blotting for RT expression. Wild-type TuYV RT protein has a theoretical molecular weight of 74.0 kDa but migrates at ~95 kDa and sometimes as a doublet, as in Figure 2B (Figure 2A,B and [16,17]). The mutant RT proteins RT-NL and RT-N65-NL have theoretical molecular weights of 74.2 kDa and 73.9 kDa, respectively. Both proteins were detected in agro-infiltrated *M. perfoliata* and *N. benthamiana* leaves (Figure 2A,B). They migrated a little bit slower than expected but in a range similar to wild-type RT. Interestingly, RT-N65-NL had similar mobility to the heavier form of the RT, whereas RT-NL migrated like the lighter RT form. This indicated that the aberrant migration of the RT protein is an intrinsic property of the RT proteins and is not due to plant-specific effects. The bands of the recombinant RT proteins in leaves agro-infiltrated with TuYV-N65-NL and TuYV-NL were considerably weaker compared to those of RT in leaves agro-infiltrated with wild-type virus. Since the deleted sequence in the RT mutants is not involved in the readthrough mechanism [22], the lower accumulation of the recombinant RT proteins is likely due to the lower stability of the proteins, although a less efficient recognition of the mutated RT proteins by the antiserum used cannot be excluded. Additional lower-weight immunoreactive bands were detected in extracts from leaves agro-infected with wild-type TuYV, which correspond to different processed forms of RT. These RT cleavage products were not visible in extracts from TuYV mutants, with the exception of a strong immunoreactive band at around 65 kDa in *N. benthamiana* agro-infiltrated with TuYV-N65-NL (Figure 2B). This additional band was less visible in TuYV-N65-NL agro-infiltrated *M. perfoliata,* and not detectible in *M. perfoliata* and *N. benthamiana* inoculated with wild-type TuYV or TuYV-NL (Figure 2A,B). The ~65 kDa band likely corresponds to an RT cleavage product caused by the replacement of the C-terminus of RT by the NL sequences.

Viral particles composed of CP and RT* (a C-terminal truncated form of the RT protein—see Figure 1) are the entities used by poleroviruses to reach distal tissues [23]. Therefore, viral particle production was analyzed by DAS-ELISA with a conformation-specific antibody recognizing virions but not free CP protein [23] and using exemplarily agro-inoculated *N. benthamiana*. Results show that virus particles were detected in wild-type-infected plants as well as those infected with the mutants (Table 1). The ELISA values were similar for all conditions, indicating equivalent virion levels in plants infected with wild-type or virus mutants. This is in line with the Western blot results using anti-CP that showed similar CP accumulation for wild-type or virus mutants (Figure 2). Having established that the mutants produced virus particles, we further analyzed the ability of the TuYV mutants to move systemically.

Systemic propagation of TuYV mutants after agro-infiltration was analyzed in *M. perfoliata*, *N. benthamiana*, and *Arabidopsis thaliana* by DAS-ELISA (Table 2). Rates of systemic infection of *A. thaliana* were clearly reduced for both mutants. Indeed, depending on the experiment, 0–50% of agro-infiltrated plants became systemically infected with the virus mutants, while infection rates of 70–100% were observed for wild-type TuYV. Viral accumulation, estimated by DAS-ELISA, was also reduced in plants infected by the mutants compared to the wild-type virus (Table 2). In *M. perfoliata*, the replacement of C-terminal RT sequences by NL or N65-NL sequences seemed to have some effect on systemic virus infection. In the two experiments, infection rates with TuYV-N65-NL (40–50%) were somewhat lower compared to wild-type TuYV (60–80%), and for TuYV-NL, infection rates were higher (90–100%) than those observed for wild-type TuYV (57–60%). Virus accumulation was not affected by the mutants in this plant species, as measured by DAS-ELISA (Table 2). For *N. benthamiana*, infection rates of TuYV-NL (90–100%) and TuYV-N65-NL (90–100%) were similar to infection rates of the wild-type virus (100%), and insertions did not affect virus accumulation compared to wild-type TuYV (Table 2). In addition, the similar accumulation of wild-type and recombinant viruses in systemically infected *M. perfoliata* and *N. benthamiana* leaves indicates that virus movement was not impeded in the mutant viruses.

The stability of the inserted sequences in the viral progeny present in systemically infected tissues was then analyzed by RT-PCR. For this, total RNA extracted from non-inoculated leaves of agro-inoculated plants was reverse-transcribed, and then the region comprising the NL or N65-NL substitution was amplified by PCR with primers specific for the insertion. Results (Figure 3) showed that the NL and the N65-NL sequences in the viral genome of the mutants were perfectly stable in *N. benthamiana* and *M*. *perfoliata* but very unstable in *A. thaliana*, where deletions of the substituted region were observed, or no PCR amplicon was obtained, suggesting major internal rearrangements of the RT sequence. The instability of recombinant sequences in *A. thaliana* but not in *N. benthamiana* and *M. perfoliata* has been reported before for the TuYV-RT_GFP_ infectious clone where the C-terminal part of the RT was replaced by GFP [19].

Because inoculation of *N. benthamiana* yielded high infection rates, a better reproducibility of results (compare experiments 1 and 2 in Table 1) and transgenesis is well established for this plant species, and further experiments were pursued in this plant species.

### 2.2. Functional Characterization: Bioluminescence Observation in Nicotiana Benthamiana

We first evaluated the detection of NanoLuc-tagged viruses by luminescence in wild-type *N. benthamiana*. For this, we agro-infiltrated leaves with TuYV-NL encoding full-length NanoLuc or with TuYV-N65-NL encoding the N-terminal part of NanoLuc. In the latter case, leaves were co-infiltrated with pMDC43-C66, encoding the C-terminal part of NanoLuc, to allow luminescence by self-assembly of RT-N65-NL and C66-NL to a functional NanoLuc molecule. As a control, leaves were agro-infiltrated with plasmids pMDC43-C66 and pMDC43-N65 expressing free NL-N65 and NL-C66, respectively. Five days after infiltration, NanoLuc substrate was infiltrated in the leaves, and luminescence was recorded. As shown in Figure 4A, we observed bioluminescence in leaves agro-infiltrated with TuYV-NL alone, co-infiltrated with TuYV-N65-NL and C66-NL, and for the positive control (free N65-NL and C66-NL). In all cases, luminescence was observed in the entire infiltrated zone, indicating that agro-infiltration induced viral replication in all infiltrated cells. This is in line with previous reports showing viral expression in agro-infiltrated zones [18,19]. The intensity of the luminescence signal was comparable for the control and for TuYV-NL and somewhat weaker for TuYV-N65-NL co-infiltrated with C66-NL. Thus, the results indicated that both split NanoLuc and full-length NanoLuc can be visualized in agro-infiltrated *N. benthamiana* leaves and that viral infection can be followed by this technique.

We next evaluated the systemic movement of the virus mutants in *N. benthamiana*. For this, we used only TuYV-NL to avoid agro-infiltration of upper leaves with C66-NL, required for reconstitution of a functional NanoLuc in the case of infection with TuYV-N65-NL. Three weeks after agro-infiltration, upper non-inoculated leaves were either detached and infiltrated with NanoLuc substrate (Figure 4B), or the substrate was infiltrated in a leaf on an intact plant (Figure 4C) and screened for luminescence. Figure 4B,C shows luminescence in major and minor veins of the infiltrated zone, as expected for a phloem-restricted virus. This confirms the RT-PCR results (Figure 3) and shows again that the NanoLuc-tagged virus is able to move over long distances and infect plants systemically.

To detect the bioluminescence of self-assembled split NanoLuc, transgenic *N. benthamiana* plants expressing C66-NL under the control of 35S CaMV promoter were constructed. The T-DNA insertion in transgenic plants was controlled by PCR before each experiment (see Appendix A). Two independent lines (1 and 3) were agro-infiltrated with TuYV-N65-NL and analyzed by DAS-ELISA, RT-PCR, and bioluminescence. DAS-ELISA (Table 3) shows that TuYV-N65-NL infected the transgenic plants similarly to wild-type *N. benthamiana* (compare Table 2 and Table 3). The infection rates were slightly lower for the N65-NL-tagged virus when compared to the wild-type virus, but viral accumulation was similar for the wild-type virus and TuYV-N65-NL (Table 3). No difference was detected in infectivity and viral accumulation between the two transgenic lines (Table 3). RT-PCR of randomly chosen ELISA-positive plants indicated stable maintenance of the viral genome region containing the N65-NL substitution in four out of five plants from line 1 and in all the eight tested plants from line 3 (Figure 5).

Taken together, the C66-NL transgene did not (line 3) or only marginally (line 1) compromise the infectivity of TuYV-N65-NL in *N. benthamiana* plants. Next, plants from lines 1 and 3 were agro-infiltrated with TuYV-N65-NL. Three weeks later, systemically infected upper leaves were infiltrated with NanoLuc substrate. As shown in Figure 6, a strong luminescence signal was observed in infected line 3 plants and a barely visible signal in infected plants from line 1. We assume that this is likely due to a higher expression level of C66-NL in line 3 plants compared to line 1 plants and not to a lower accumulation of TuYV-N65-NL in these plants since the virus accumulated to similar levels in both lines (Table 3). Please note that in this experiment, we were not able to infiltrate the whole leaf area with the substrate; therefore, the luminescence pattern was patchy. To conclude, our results show that luminescence can be used to follow TuYV infection in wild-type and transgenic *N. benthamiana* plants, either using full-length or split NanoLuc.

## 3. Discussion

In this study, we evaluated the use of NanoLuc and split NanoLuc as a bioluminescence reporter system to follow the infection of plants with a plant virus: turnip yellows virus. We chose a substitution strategy to introduce the reporters into the TuYV genome and replaced the C-terminal 171 or 65 amino acids of the RT protein with full-length NanoLuc or N65-NL split NanoLuc sequences, respectively. This resulted in RT:NanoLuc fusion proteins and conserved the total length of the TuYV genome. We used this strategy to ensure the encapsidation of the recombinant genomes into the icosahedral capsid that offers only limited capacities to accommodate additional genomic material. We previously used this approach successfully to introduce a GFP-based reporter system into TuYV [19]. In both cases, infectious viruses maintained the reporter sequence stably in *N. benthamiana* and *M. perfoliata* but not in *A. thaliana,* where the exogenous sequences were rapidly deleted. We have no precise explanation for this. We assume that the C-terminus of RT may be required for infectivity in *A. thaliana* but not in the two other hosts. A host-specific effect of the C-terminus of the RT protein has been reported before since the deletion of the C-terminal 200 amino acids compromised TuYV infectivity in *N. benthamiana*, but less so in *A. thaliana* and *M. perfoliata* [18]. The discrepancy in the results could be due to the length of the deletion introduced in the RT, which was smaller in our study compared to the previous one (171 and 65 amino acids compared to 200 amino acids). Whatever the case, our results show that TuYV-NL and TuYV-N65-NL can reliably infect at least two plant species and maintain the inserted NanoLuc sequences, which is a pre-condition for the use of this reporter system. In our previous work using GFP-tagged TuYV, a GFP signal was observed in agro-inoculated *N. benthamiana* leaves. Like in the present work, the labeling was visible in the entire agro-infiltrated area. In contrast, the GFP label was restricted to the phloem in systemically infected leaves, and the veins were not uniformly labeled. Rather the GFP fluorescence was patchy along the veins, indicating that not all phloem cells were infected. We observed here a very similar distribution of TuYV bioluminescence in systemically infected leaves, with the signal being irregular and restricted to the phloem as well. The GFP label seemed to be weaker than the bioluminescence signal in both agro-infiltrated and systemically infected leaves (compare Figure 4 in our previous publication with Figure 4 in this work). However, it is extremely difficult to compare the sensitivity of the two labeling methods, as it depends enormously on the acquisition conditions such as the distance of observation, camera equipment and lenses, objectives used, filter settings, and so on.

To our knowledge, this is the first time that NanoLuc and split NanoLuc was applied to a plant virus. TuYV could be detected in tissues with both reporters in initially agro-infiltrated regions of the leaf or exclusively in the phloem in upper, systemically infected leaves, as expected for a phloem-limited virus. Therefore, real-time observation of TuYV infection was possible, both in detached leaves and in intact plants. One drawback was that the NanoLuc technique requires—like other luciferase systems—the addition of luciferase substrate before observation. This can be a delicate step, especially for older leaves and veins that are more difficult to infiltrate. A second inconvenience of the technique is the rather small time window for observation (1–2 h) after substrate addition. However, the recent development of time-released substrates overcomes this problem and allows for much longer observation times [24]. Comparing sensitivity, the bioluminescence signal was slightly weaker for the split NanoLuc configuration compared to full-length NanoLuc. This is expected because the self-assembly efficiency of N65-NL and C66-NL is not 100% [14], lowering the concentration of reconstituted active NanoLuc from N65-NL/C66-NL mixtures compared to full-length NanoLuc.

Taken together, we demonstrate here the feasibility of the NanoLuc system for the real-time detection of a plant virus. It might represent, in some circumstances, an alternative to the widely used fluorescent protein reporter systems because the bioluminescence signals are easier to quantify than fluorescence. It can be—due to its high signal-to-noise ratio—more sensitive, and it allows observation deep in the plant tissue. Not negligible is also the possibility of measuring bioluminescence in plant extracts, which might facilitate measuring virus titers. Finally, the full-length NanoLuc (19 kDa) molecule is smaller than most fluorescent proteins (25–35 kDa for monomeric fluorescent proteins), which is advantageous if there are space constraints, for example, limited capsid capacity to accommodate additional nucleic sequences. Also, the fusion of a smaller reporter to a protein decreases the chances that the fusion changes protein function by steric hindrance or by causing protein misfolding. In this regard, the split NanoLuc provides an additional advantage with the reduced size of the inserted fragment (195 bp and 7.2 kDa for split NanoLuc vs. 513 bp and 19 kDa for NanoLuc). Recently, another split NanoLuc system was developed that uses an even smaller fragment of ~30 bp translated into a 1.3 kDa protein that self-assembles with high affinity with an 18 kDa moiety of NanoLuc to reconstitute bioluminescence [25]. Taken together, this work represents a step forward in the use of bioluminescence to follow virus infection in real-time in a non-disruptive manner and might increase the attractiveness of this reporter system.

## 4. Materials and Methods

### 4.1. Cloning of TuYV-N65-NL and TuYV-NL

In TuYV-N65-NL, for split NanoLuc detection, nucleotides 5298 through 5493 of the TuYV sequence coding for the C-terminal 65 amino acids of RT were replaced by the 65 N-terminal amino-acids from NanoLuc (N65-NL). For this, the N65-NL sequence was amplified by PCR from pET.N65 plasmid template [14] with TaqPhusion polymerase (Thermofisher Scientific, Courtaboeuf, France), RP1 and FP3 primers (Appendix A) containing TuYV sequence extensions (234 bp in total), using the amplification protocol: 98 °C 30 s; 5 cycles of 98 °C 10 s, 54 °C 20 s, 72 °C 30 s, followed by 25 cycles of 98 °C 10 s, 68 °C 20 s, 72 °C 30 s and final extension for 10 min at 75 °C (PCR1). pBS.TuYV-G∆RN [15] plasmid was used as a template to amplify the plasmid and TuYV sequence (without the C-terminal sequence of RT protein) with TaqPhusion polymerase and FP2 and RP3 primers containing N65-NL sequence extensions (PCR 2). FP2 and RP3 primers are totally complementary to RP1 and FP3 primers, respectively, used in PCR1. PCR2 amplification was performed using the following steps: 98 °C 30 s, 5 cycles of 98 °C 10 s, 54 °C 20 s, 72 °C 3 min 20 s; 30 cycles of 98 °C 10 s; 68 °C 20 s; 72 °C 3 min 20 s and final extension at 75 °C for 10 min. Plasmid templates from PCR1 and 2 were removed by a *Dpn*I treatment for 1 h at 37 °C (NEB, Evry, France), and PCR products were purified in NucleoSpin Gel and PCR clean-up columns (Macherey-Nagel, Hoerdt, France) following the instructions of the manufacturer. A Gibson reaction was performed according to the manufacturer’s protocol (NEB, Gibson Assembly Master Mix) for 15 min at 50 °C in a final volume of 20 µL using 13 ng and 50 ng of PCR1 and PCR2 products, respectively. This recombination led to the substitution of the 65 last amino acids of RT protein by the 65 N-terminal amino acids of NanoLuc.

For the cloning of TuYV-NL, where the C-terminal 171 amino acids of RT (nucleotides 4980–5493 of the TuYV genome) are replaced by NanoLuc, the complete NanoLuc sequence was amplified by PCR from the pNL1.1 plasmid template (Promega, Charbonnières-Les-Bains, France) using TaqPhusion polymerase and RP4 and FP4 primers containing TuYV sequence extensions (556 bp) (PCR3). pBS.TuYV-G∆RN plasmid was used as a template to amplify the plasmid and TuYV sequence (without the C-terminal sequence of RT protein) (PCR4) using TaqPhusion polymerase and FP5 and RP5 primers containing NanoLuc sequence extensions. FP5 and RP5 primers are complementary to RP4 and FP4 primers, respectively, which were used in PCR3. PCR amplifications were performed using this protocol: 98 °C 30 s; 5 cycles of 98 °C 10 s, 54 °C 20 s, 72 °C 3 min 20 s; 30 cycles of 98 °C 10 s, 68 °C 20 s, 72 °C 3 min 20 s; final extension at 75 °C for 10 min. Plasmid templates from PCR 3 and 4 were removed by *Dpn*I treatment, and PCR products were column-purified. A Gibson reaction was performed for 15 min at 50 °C according to the manufacturer’s protocol (NEB) using 8 ng of PCR3 product and 30 ng of PCR4 product in a final volume of 20 µL. This reaction led to the exchange of the last 171 amino acids of the RT protein by the complete NanoLuc sequence (171 amino acids, [13]).

Then, for both TuYV-N65-NL and TuYV-NL, 5 µL of the Gibson reaction was used to transform XL10-Gold thermocompetent cells. Recombinant cells were selected on LB ampicillin medium, and Gibson recombination was analyzed by PCR using specific luciferase and TuYV primers. Recombinant clones were sequenced from *Spe*I to *Sal*I restriction sites (nucleotides 1350–5641 of wild type TuYV in pBS.TuYV-G∆RN sequence), and for each construct, a positive clone was digested with *Spe*I and *Sal*I restriction enzymes to release most of the virus sequence including the inserted NanoLuc sequences. An additional digestion with the *Nde*I enzyme was performed to cut the pBS plasmid backbone and to facilitate recovery of the *Spe*I–*Sal*I fragment from an agarose gel. The purified *Spe*I–*Sal*I fragment was inserted into pBinTuYV-G∆RN (Leiser et al., 1992) and linearized with the same enzymes (*Spe*I–*Sal*I) to obtain the binary vectors pBin-TuYV-N65-NL and pBin-TuYV-NL.

For agro-inoculation, pBin-TuYV-N65-NL and pBin-TuYV-NL were introduced into *Agrobacterium tumefaciens* strain C58C1 by electroporation according to the protocol described [26]. A binary vector containing the complete sequence of wild-type TuYV virus (pGΔRN, [15]) was introduced into *A. tumefaciens* strain C58C1 was used as a positive control in infection experiments.

### 4.2. Cloning of C66-NanoLuc Domain in Binary Vector for Agro-Infiltration and Plant Transformation

To obtain the binary vector pMDC43-C66 expressing the C-terminal 106 amino acids of NanoLuc (C66-NL) under the control of the 35S promoter, the C66-NL sequence and a C-terminal myc-tag were cloned into the pMDC43 plasmid [27]. For this, the C66-NL sequence was PCR-amplified using pRSF-66C plasmid as a template [14] with primers Luc-For (Appendix A) containing an additional *Xba*I site upstream of the C66-NL start codon and Luc-Rev containing an additional linker and a myc-tag sequence before the C66-NL stop codon and a downstream *Sac*I site. After PCR (2 min denaturation at 95 °C, 34 cycles of 30 s 95 °C, 20 s 52 °C, 35 s 72 °C, final prolongation 5 min 72 °C), the amplicon was purified and digested with *Xba*I and *Sac*I and ligated into gel-purified pMDC43 cut with the same restriction enzymes. Thermocompetent DH5α cells were transformed with the ligation product, recombinant clones identified by colony PCR, and inserts verified by sequencing. For agro-infiltration and plant transformation, pMDC43-C66 was introduced into electrocompetent *A. tumefaciens* LBA4404 cells.

### 4.3. Nicotiana Benthamiana Transformation

Sterilized leaves of *N. benthamiana* were cut into 1 cm^2^ squares, and leaf segments were immersed in an *A. tumefaciens* suspension (OD_600nm_ 0.8) transformed with pMDC43-C66 and incised with a scalpel blade. Leaf segments were placed in a growth chamber (16 h light, 24 °C/8 h dark, 22 °C) onto Murashige and Skoog medium-containing agarose ((MS, Duchefa, Haarlem, The Netherlands), 3% *w*/*v* sucrose, 0.05 μg/mL 1-naphthalene acetic acid, 2 μg/mL 6-benzyl-aminopurine, and 0.65% *w*/*v* agar, pH 5.8). After 4 days, leaf segments were transferred onto a new culture medium enriched with 25 µg/mL hygromycin and 500 μg/mL carbenicillin and were cultivated on this medium until plantlets could be acclimatized in soil. The T-DNA insertion in the plant genome was confirmed by PCR using Kapa3G plant PCR kit (Clinisciences, Nanterre, France) and primers 947 and 1048 (Appendix A). After self-pollination, experiments were performed on T2 generation.

### 4.4. Plant Growth and Inoculation

Wild-type and transgenic *Nicotiana benthamiana*, *Arabidopsis thaliana*, and *Montia perfoliata* plants were cultivated in a growth chamber in TS 3 fine substrate (Klasmann-Deilmann, Bourgoin Jallieu, France) under 22 ± 1 °C and 14 h photoperiod under LED lights. For agro-inoculation, *A. tumefaciens* cell cultures were grown to an OD_600nm_ of 0.5 before being agro-infiltrated with a syringe into three to five-week-old plants as described [19,28] and maintained under the same growth conditions until analysis.

### 4.5. Western Blot

CP and RT protein synthesis by virus mutants and wild-type TuYV were analyzed by western blot on agro-infiltrated leaves 5 days after inoculation, using polyclonal antisera raised against TuYV-CP [26] and TuYV-RT proteins [29]. Briefly, 100 mg of leaves were sampled and ground in liquid nitrogen. The powder was resuspended in 400 µL of 5X Laemmli buffer [30]. Boiled samples (15 min at 90 °C) were submitted to sodium dodecyl sulfate-polyacrylamide gel electrophoresis (SDS-PAGE), and separated proteins were transferred to a nitrocellulose membrane. The membrane was incubated with primary antibodies directed against CP and RT proteins and with a secondary antibody coupled to horseradish peroxidase (goat anti-rabbit-HRP, ThermoFisher). After the addition of horseradish peroxidase substrate (Lumi-LightPLUS kit; Roche, Meylan, France), antibody-antigen complexes were detected with a cooled high sensitivity camera (Syngene Gbox, Fisher Scientific, Illkirch, France).

### 4.6. Virus Detection by DAS-ELISA

Virus detection in plant tissue was performed by double-antibody sandwich ELISA on agro-infiltrated and systemic leaves, as described previously [17]. TuYV-specific antiserum (Loewe, Sauerlach, Germany) was used at a dilution of 1/400 (*v*/*v*). Infection rates were compared to wild-type viruses in different host plants using a binomial generalized linear model (GLM) on R software version 4.2.1 (https://www.r-project.org/ (accessed on 12 January 2022)).

### 4.7. Analysis of Viral Progeny by RT-PCR

Detection of viral RNA was performed by RT-PCR on total RNA extracted from 100 mg of fresh plant samples using a commercial RNA purification kit (RNeasy Plant Mini Kit; Qiagen, Courtaboeuf, France) and according to the provider’s instructions. RNA was eluted in 30 µL of RNase-free water, and reverse transcription was performed with M-MLV reverse transcriptase (Promega) on 10 µL of RNA and with primer 221, which is complementary to 3′ UTR sequence of the TuYV recombinants. The complementary DNA obtained was used as a template for PCR amplification with forward primer (946), which hybridized at the very beginning of the NanoLuc sequence and the reverse primer 221. After 30 cycles of amplification with GoTaq polymerase (Promega), the amplified cDNA sequences were visualized on agarose gel in 0.5X Tris/acetate/EDTA buffer after ethidium bromide staining.

### 4.8. Virus Detection by Bioluminescence

Prior to observation, NanoLuc substrate (Nano-Glo Live Cell Assay System N2011, Promega) was mixed with the provided substrate buffer, diluted in 10 mM MgCl2, 5 mM MES pH 6.5, and infiltrated into leaves. Luminescence was observed with a cooled high-sensitivity camera (Syngene Gbox).

## Figures and Tables

**Figure 1 ijms-23-13685-f001:**
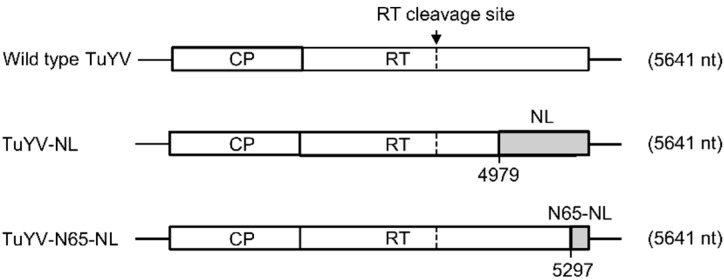
Schematic diagram showing insertion of the complete NanoLuc (NL) sequence or its N-terminal domain (N65-NL) (grey boxes) into the 3′ part of the TuYV genome. The C-terminal 171 (for NL) or 65 (for N65-NL) amino acids of the readthrough protein were replaced by NL or N65-NL sequence leading to TuYV-NL and TuYV-N65-NL mutants having the same genome size as wild-type TuYV (5641 nt). White boxes represent open reading frames coding for coat protein (CP) and readthrough protein (RT). The black arrow indicates the RT cleavage site at amino acid 437 [20], giving rise to the truncated RT protein (RT*) that is incorporated in the capsid.

**Figure 2 ijms-23-13685-f002:**
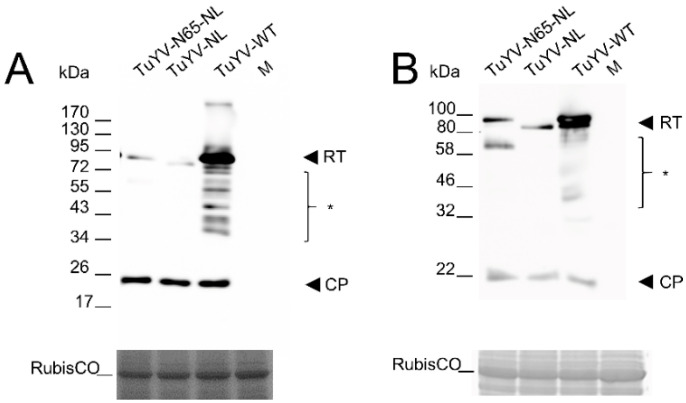
Western blot analysis of (**A**) *Montia perfoliata* and (**B**) *Nicotiana benthamiana* leaves agro-infiltrated with TuYV-N65-NL, TuYV-NL, with wild-type TuYV (TuYV-WT) or mock-inoculated (M). Two acrylamide gels were loaded in parallel with total protein extracts. Proteins from the first gel were transferred onto a nitrocellulose membrane and incubated with a mix of antibodies raised against the TuYV coat protein (CP) or the readthrough protein (RT). The second gel was stained with Instant Blue solution, and the large RubisCO unit is shown as a loading control. *, different cleavage products of the RT protein; positions of the molecular markers (in kDa) are indicated on the left.

**Figure 3 ijms-23-13685-f003:**
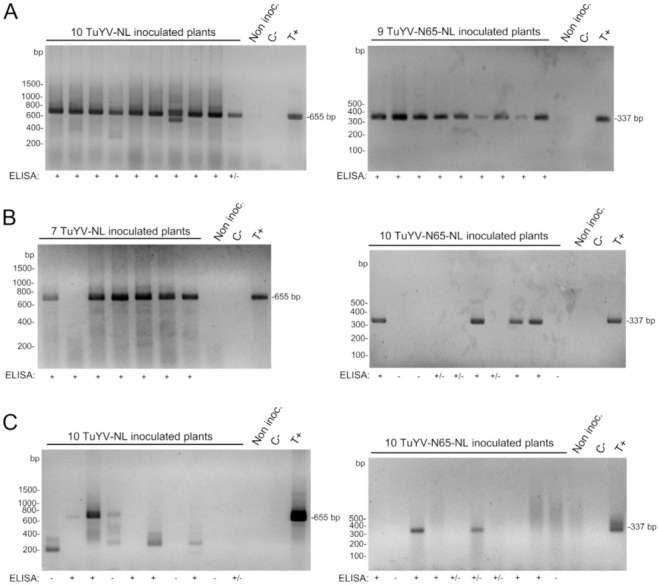
Analysis of viral progeny by reverse transcription-polymerase chain reaction (RT-PCR) in non-inoculated (systemically infected) leaves of (**A**) *Nicotiana benthamiana*, (**B**) *Montia perfoliata*, and (**C**) *Arabidopsis thaliana* agro-infiltrated with TuYV-NL or TuYV-N65-NL. Total RNA was isolated two weeks post-inoculation from plants that were positive in ELISA (+), negative in ELISA (−), or had an OD value just above the threshold value in ELISA (+/−). For PCR, primers binding to the beginning of the NanoLuc sequence and to the 3′ UTR of the TuYV genome were used. An amplicon of 655 bp was expected for TuYV-NL and 337 bp for TuYV-N65-NL. Non-inoc., non-inoculated plants; C-, non-template control, T+, positive plasmid control (pTuYV-NL for the left panel and pTuYV-N65-NL for the right panel); bp, base pairs.

**Figure 4 ijms-23-13685-f004:**
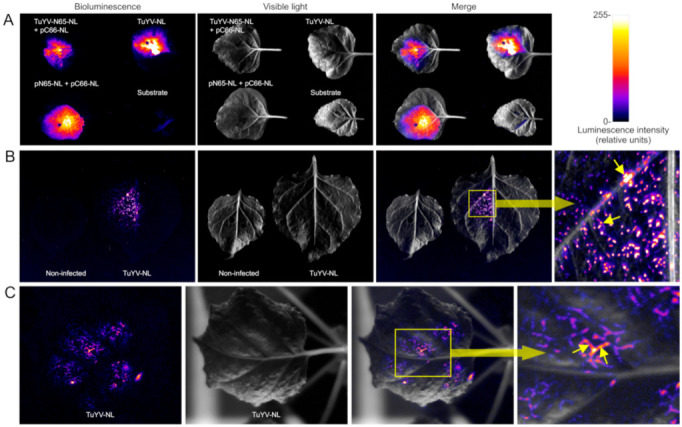
Detection of TuYV-NL and TuYV-N65-NL bioluminescence in wild-type *Nicotiana benthamiana* plants after infiltration of Nano-Glo substrate. (**A**) Nano-Glo substrate was injected in the inoculated leaf five days after agro-infiltration with TuYV-N65-NL and a plasmid expressing C66-NL from a 35S promoter (to allow the self-assembly of functional NanoLuc) or after agro-infiltration of TuYV-NL. As a positive control, leaves were agro-infiltrated with plasmids expressing N65-NL and C66-NL from 35S promoters (pN65-NL + pC66-NL), and as a negative control, Nano-Glo was infiltrated in a non-infiltrated leaf (Substrate). (**B**) Nano-Glo substrate was infiltrated in a detached upper, systemically infected leaf three weeks after agro-infiltration with TuYV-NL. A non-infected leaf was infiltrated with Nano-Glo as a negative control. (**C**) Nano-Glo substrate was infiltrated *in planta* in a systemically infected leaf three weeks after agro-infiltration with TuYV-NL. Bioluminescence or visible light was recorded in a G-box, and luminescence intensity is presented in a false color scale. The regions delineated by the yellow boxes are shown in higher magnification at the right, and the small yellow arrows point to bioluminescence in leaf veins.

**Figure 5 ijms-23-13685-f005:**
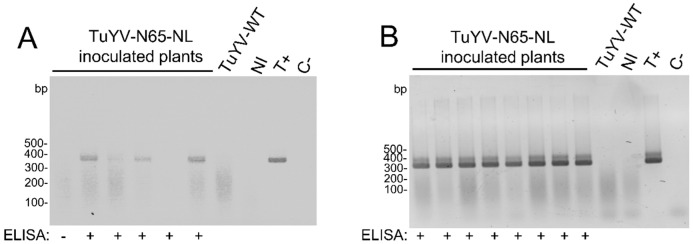
Analysis of viral progeny by reverse transcription-polymerase chain reaction (RT-PCR) in non-inoculated leaves of *Nicotiana benthamiana* 35S:C66-NL line 1 (**A**) and line 3 (**B**) agro-inoculated with TuYV-N65-NL. Total RNA was isolated two weeks post-inoculation from plants positive (+) or negative (−) in ELISA. For PCR, primers binding to the beginning of the NanoLuc sequence and to the 3′ UTR of the TuYV genome and amplifying a 337 bp fragment were used. TuYV-WT, plants inoculated with wild type TuYV; NI, non-inoculated plants; T+, positive plasmid control (pTuYV-N65-NL); C−, non-template control; bp, base pairs.

**Figure 6 ijms-23-13685-f006:**
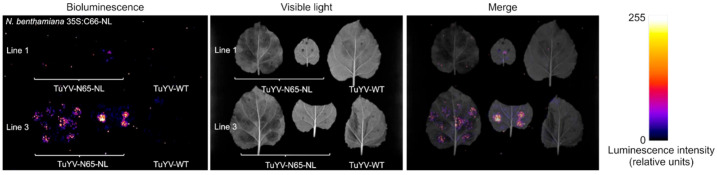
Detection of TuYV-N65-NL bioluminescence in two transgenic *Nicotiana benthamiana* lines (1 and 3) expressing C66-NL from a 35S promoter. Nano-Glo substrate was infiltrated in an upper, systemically infected leaf three weeks after agro-infiltration with TuYV-N65-NL. A leaf systemically infected with wild-type TuYV (TuYV-WT) was infiltrated with Nano-Glo as a negative control. Bioluminescence or visible light was recorded in a G-box. Luminescence intensity is presented in a false color scale.

**Table 1 ijms-23-13685-t001:** Detection by double antibody sandwich-ELISA of TuYV-N65-NL and TuYV-NL in inoculated leaves of *Nicotiana benthamiana*. Two plants were analyzed for inoculated conditions and three for non-inoculated conditions.

Virus	OD_405nm_ ± SD
TuYV-N65-NL	1.46 ± 0.21
TuYV-NL	1.55 ± 0.01
Wild type TuYV	1.47 ± 0.22
Not inoculated	0.11 ± 0.01

**Table 2 ijms-23-13685-t002:** Detection by double antibody sandwich (DAS)-ELISA of wild-type TuYV (TuYV-WT), TuYV-NL, and TuYV-N65-NL in non-inoculated (systemically infected) leaves of three plant species in different experiments (Exp.).

	*Nicotiana benthamiana*				*Montia perfoliata*				*Arabidopsis thaliana*					
	Exp. 1		Exp. 2		Exp. 1		Exp. 2		Exp. 1		Exp. 2		Exp. 3	
	Nb pl. inf./ pl. inoc. ^1^	OD ± SD ^2^	Nb pl. inf./ pl. inoc. ^1^	OD ± SD ^2^	Nb pl. inf./ pl. inoc. ^1^	OD ± SD ^2^	Nb pl. inf./ pl. inoc. ^1^	OD ± SD ^2^	Nb pl. inf./ pl. inoc. ^1^	OD ± SD ^2^	Nb pl. inf./ pl. inoc. ^1^	OD ± SD ^2^	Nb pl. inf./ pl. inoc. ^1^	OD ± SD ^2^
TuYV-NL	10/10 (100%)	0.63 ± 0.25	9/10 (90%)	0.61 ± 0.10	7/7 (100%) **	0.87 ± 0.52	9/10 (90%) **	0.70 ± 0.25	5/10 (50%) **	0.21 ± 0.09	4/10 (40%) **	0.39 ± 0.16		
TuYV-WT	8/8 (100%)	0.83 ± 0.26	8/8 (100%)	0.80 ± 0.14	4/7 (57%)	0.79 ± 0.40	6/10 (60%)	1.18 ± 0.23	10/10 (100%)	1.02 ± 0.14	8/10 (80%)	0.91 ± 0.20		
Non-inoc. ^3^	0/3	0.11 ± 0.00	0/3	0.10 ± 0.00	0/3	0.11 ± 0.00	0/1	0.11	0/3	0.05 ± 0.00	0/1	0.11		
TuYV-N65-NL	9/10 (90%)	1.05 ± 0.44	9/9 (100%)	1.35 ± 0.17	5/10 (50%)	0.99 ± 0.67	4/10 (40%)	0.99 ± 0.35	0/10 (0%) ***	** */* **	1/15 (6.7%) ***	0.28	4/10 (40%)	0.56 ± 0.41
TuYV-WT	10/10 (100%)	0.81 ± 0.16	10/10 (100%)	1.22 ± 0.28	6/10 (60%)	1.93 ± 0.41	8/10 (80%)	1.21 ± 0.20	7/10 (70%)	1.08 ± 0.62	11/15 (73.3%)	1.69 ± 1.10	7/10 (70%)	2.66 ± 0.26
Non-inoc. ^3^	0/3	0.10± 0.00	0/3	0.14 ± 0.00	0/2	0.10 ± 0.00	0/3	0.13 ± 0.01	0/2	0.10 ± 0.00	0/3	0.12 ± 0.00	0/3	0.124 ± 0.01

^1^ Number of infected plants/inoculated plants. ^2^ OD ± SD: Average OD_405nm_ value ± standard deviation (SD). Plants are considered infected when the optical density (OD) value of the leaf extract is more than two times the mean OD values of the three non-inoculated plants, plus three times the standard deviation of this mean value. Plants with OD values close to the threshold value were reanalyzed by RT-PCR for the presence/absence of the virus. ^3^ Non-inoculated plants. *, **, and *** indicate statistically significant differences in infection rates between TuYV wild-type and TuYV mutants with *p*-values < 0.05, <0.01, and <0.001, respectively (binomial GLM).

**Table 3 ijms-23-13685-t003:** Detection by double antibody sandwich (DAS)-ELISA of TuYV-N65-NL in upper non-inoculated leaves of two lines (1 and 3) of transgenic *Nicotiana benthamiana* 35S:C66-NL.

	Line 1		Line 3	
	Nb pl. inf./ pl. inoc. ^1^	OD ± SD ^2^	Nb pl. inf./ pl. inoc. ^1^	OD ± SD ^2^
ocTuYV-N65-NL	7/10 (70%)	0.92 ± 0.34	9/10 (90%)	0.82 ± 0.35
TuYV-WT	10/10 (100%)	0.99 ± 0.20	10/10 (100%)	0.88 ± 0.15
Non-inoculated	0/3	0.10± 0.00		

^1^ Number of infected plants/inoculated plants. Plants are considered infected when the optical density (OD) value of the leaf extract is more than twice the mean OD values of the three non-inoculated plants plus three times the standard deviation of this mean value. ^2^ OD ± SD, Average OD_405nm_ of infected plants ± standard deviation.

## Data Availability

The data presented in this study are available on request from the corresponding author.

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
