# Peer review of "Bioluminescence Production by Turnip Yellows Virus Infectious Clones: A New Way to Monitor Plant Virus Infection"

_ijms, 2022, doi:10.3390/ijms232213685_

Round 1

Reviewer 1 Report

In present study, the authors used the NanoLuc luciferase bioluminescent reporter system to detect turnip yellows virus (TuYV) in infected plants. The authors found thatBioluminescence imaging detected TuYV-NL in inoculated and systemically infected leaves in Nicotiana benthamiana. And then, the authors further constructed transgenic N. benthamiana plants expressing the C-terminal domain of split NanoLuc, and found Bioluminescence imaging of these plants after agro-infiltration with TuYV-N65-NL allowed detection of virus in systemically infected leaves. This study is interesting and contributes to our understanding of monitor plant virus infection. And the present MS need minor revision before the manuscript could be accepted for publication.

1.       Please give more background of turnip yellows virus.

2.       Table 1-3, the font format is inconsistent with the font in the main text.

3.       Figure 5B, why two band were showed in this figure? Clear one band were showed in figure 3.

4.       Figure 6, Detection of TuYV-N65-NL bioluminescence in two transgenic Nicotiana benthamiana lines (A and C) expressing C66-NL from a 35S promoter. I can not find A and C in this figure. 

Reviewer 2 Report

The ms is nicely written and the word described clearly. This reviewer finds that this ms should be improved by incorporating the following points:

1. Since this ms describes a new method, it is necessary to compare the results of detection of the labeled virus described here with that of earlier reports, which used fluorescent proteins or other similar systems with respect to sensitivity. 

2. Most viruses are expected to move systemically along the vascular bundles. In that case, signals should have been observed in the veins. Figures 4 and 6 do not show any significant signals from the veins. The authors should discuss this aspect based on the current knowledge.

3. The engineered viruses described here appear to be no different in accumulation levels as compared with the wild type virus (Figure 2 and Table 2). Is it also true for movement to systemic tissues? The authors should address that question as well.

Reviewer 3 Report

A meaningful study is described in this manuscript. Virus infection in real time and in a non-destructive manner is desirable for various reasons. In contrast, the NanoLuc luciferase bioluminescence reporter system has more potential to detect turnip yellows virus(TuYV) infected plants. TuYV was genetically tagged by replacing the C-terminal part of the RT protein with full length NanoLuc (TuYV-NL) or with the N-terminal domain of split NanoLuc (TuYV-N65-NL). Wild type and recombinant viruses were agro-infiltrated in Nicotiana benthamiana, Montia perfoliata and Arabidopsis thaliana. Western blot, DAS-ELISA, RT-PCR and Bioluminescence imaging experiments confirmed that the NanoLuc luciferase can be used to monitor infection with TuYV. In general, this work is sufficient and the conclusion is reliable. I suggest that this article could be accepted for publication after minor revision as the following points:

1. Agro-infiltrated plant leaves with the wild type and mutant constructs to assess infectivity of the virus mutants. But only for Montia perfoliata and Nicotiana benthamiana, why not Arabidopsis thaliana ?

2. The Bioluminescence observation experiment only in Nicotiana benthamiana. It is recommended that the experiment be carried out for all three species mentioned in this paper.

3. line 353, “72° C” would better changed to “72 °C”. Others similary modify.

4. The References 16, line 530, would better add the DOI number. Others similary modify.

Reviewer 4 Report

Boissinot et al submitted a manuscript titled "Bioluminescence production by turnip yellows virus infectious clones: a new way to monitor plant virus infection" for publication in IJMS, however it doesn't fit in the scope of this journal as this technique of fluorescent tagging a plant virus is very common, and extensively done already in 10s of viruses, if not more. This work as such has very less value in pushing the field of molecular sciences any further. However, the authors are suggested to submit this work in Virology subject journals, which are much more relevant and better monitored.

Round 2

Reviewer 4 Report

"Bioluminescence production by turnip yellows virus infectious clones: a new way to monitor plant virus infection" submitted by Boissinot et al doesn't still qualify for publishing in IJMS. As suggested earlier, it is more suitable for publication in Virology or plant journals.